# A Novel *GBF1* Variant in a Charcot-Marie-Tooth Type 2: Insights from Familial Analysis

**DOI:** 10.3390/genes15121556

**Published:** 2024-11-29

**Authors:** Valentina Ciampana, Lucia Corrado, Luca Magistrelli, Elena Contaldi, Cristoforo Comi, Sandra D’Alfonso, Domizia Vecchio

**Affiliations:** 1Neurology Unit, Department of Translational Medicine, Maggiore Della Carità Hospital, University of Piemonte Orientale, 28100 Novara, Italy; 20022116@studenti.uniupo.it (V.C.); cristoforo.comi@med.uniupo.it (C.C.); 2Department of Translational Medicine, Genetic Laboratory, Università del Piemonte Orientale (UPO), 28100 Novara, Italy; lucia.corrado@med.uniupo.it; 3Department of Health Sciences, Center on Autoimmune and Allergic Diseases (CAAD), University of Piemonte Orientale (UPO), 28100 Novara, Italy; sandra.dalfonso@med.uniupo.it; 4Parkinson Institute of Milan, ASST G.Pini-CTO, 20122 Milano, Italy; magis.luca@gmail.com (L.M.); contaldie@yahoo.it (E.C.)

**Keywords:** Charcot–Marie–Tooth type 2 (CMT2), hereditary motor/sensory peripheral neuropathy, Golgi brefeldin A resistant guanine nucleotide exchange factor 1 (*GBF1*)

## Abstract

Background/Objectives: Axonal Charcot–Marie–Tooth disease type 2 (CMT2) accounts for 24% of Hereditary Motor/Sensory Peripheral Neuropathies. CMT2 type GG, due to four distinct heterozygous mutations in the Golgi brefeldin A resistant guanine nucleotide exchange factor 1 (*GBF1*) gene (OMIM 606483), was described in seven cases from four unrelated families with autosomal dominant inheritance. It is characterized by slowly progressive distal muscle weakness and atrophy, primarily affecting the lower limbs. Here, we present two siblings sharing a novel *GBF1* variant. Methods: Patient II.1 (male, 61 years at onset) presented lower limb hypoesthesia and walking difficulty; the examination revealed a postural tremor, a positive Romberg test, and muscle atrophy in the lower limbs and hands. Patient II.2 (his sister, 59 years at onset) had lower limb dysesthesias, hand paresthesia, and lower-limb stiffness. They underwent clinical evaluations, blood tests, and electroneurography. Their father represents a potentially affected individual, although a genetic analysis was not conducted. Results: All tests for peripheral neuropathies were unremarkable, including metabolic and autoimmune screening. Both showed a mixed demyelinating–axonal sensory–motor neuropathy. Genetic analysis revealed a new heterozygous *GBF1* variant of uncertain significance. Conclusions: Based on autosomal dominant inheritance, as well as clinical and physiological features, a possible novel CMT2GG was diagnosed. Further research, including functional assays and in vitro studies, is necessary to confirm this variant’s causal link.

## 1. Introduction

Charcot–Marie–Tooth disease (CMT) is the most common inherited motor and sensory neuropathy, exhibiting high clinical and genetic heterogeneity. To date, seven forms of CMT have been described, including axonal and demyelinating types, with autosomal dominant, recessive, and X-linked inheritance.

Two main subgroups are traditionally distinguished on the basis of upper limb motor nerve conduction velocity (NCV): demyelinating forms, in which motor NCV is less than 35 m/s, and axonal forms in which motor NCV is more than 45 m/s; “intermediate” conduction velocities are between 35 and 45 m/s [1]. According to the Italian Registry [2], one-quarter of CMT-patients have an axonal form, in contrast with the two-thirds with a demyelinating neuropathy and the remaining part with an intermediate type and/or mutation(s) in known CMT genes.

Initially, specific diseases associated with particular genes were assigned letters; nowadays, the large number of identified genes, thanks to Next-Generation Sequencing (NGS) techniques, has challenged the current classification of CMT [3], with over 100 different genes identified and ongoing discoveries [4]. Described pathogenic variants include whole-gene duplications, deletions, and point mutations [5], affecting genes expressed in myelin, gap junctions, ion channel proteins and/or axonal structures within peripheral nerves (e.g., motor proteins and axonal transport, tRNA syntetases) [6]. The American College of Medical Genetics and Genomics (ACMG) proposed guidelines for evaluating the pathogenicity of these newly discovered genetic variants; these criteria have been updated and improved for specific pathologies, including CMT [7]. CMT is genetically determined through a monogenic pathway; therefore, a detailed family history indicative of symptoms suggestive of neuropathy is essential to exclude the risk of the condition in patients who do not present typical characteristics of an acquired disorder.

CMT type 2 (CMT2) belongs to the axonal forms. These forms are quite common, with a reported prevalence of 12–36% [8], and tend to occur in the second or third decade of life. Clinically, patients present slowly progressive distal muscle weakness and atrophy, primarily affecting the lower limbs, causing difficulty in walking, musculoskeletal deformities, and often leading to high-arched feet (pes cavus). On the contrary, upper limbs are rarely involved. Because CMT2 progresses gradually, it is essential to use scales that can document small changes over time [9]; a useful tool to monitor progression is the CMT2 Neuropathy Score (Version 2), which considers the neuropathy severity and impact on daily activities. From a genetic point of view, they can be classified into autosomal dominant (AD) inheritance (CMT2 properly called) and autosomal recessive forms (AR-CMT2) [1]. Clinical heterogeneity is not well understood and may be due to interactions between genetic defects, genetic susceptibility factors, and environmental contributors [10]. In fact, over 60 genes, expressed in peripheral neurons and, thus, leading to axonal degeneration, have been implicated in CMT2, expanding the NGS diagnostic panel, even if in about half of affected individuals the genetic variants still remain unknown. Given the complexity of the old classification of CMT diseases, which are non-uniform and constantly evolving thanks to the growing discovery of new genetic variants, a new version has been proposed, based on the pattern of inheritance, the description of the phenotype, and the name of the gene(s) implicated [11]. However, there is still no uniformity in the current clinical practice [12].

CMT type 2GG (CMT2GG) is a newly described adult-onset AD axonal peripheral neuropathy (OMIM 606483). Mendoza-Ferreira et al. [13] reported seven patients from four unrelated families with axonal peripheral neuropathy, identifying different heterozygous mutations (three missense and one nonsense variant) in the Golgi brefeldin A resistant guanine nucleotide exchange factor 1 (*GBF1*) gene, which is potentially inherited in an AD manner with incomplete penetrance. This is a housekeeping gene that is ubiquitously expressed in over 25 tissues [14]; its expression in the brain (particularly in the nucleus accumbens and putamen of the basal ganglia and substantia nigra), tibial nerve, and spinal cord are of particular relevance in this paper. The encoded protein of 1860 amino acids is localized predominantly in the Golgi apparatus. *GBF1* is required for Golgi apparatus assembly, for vesicular trafficking by activating ADP ribosylation factor 1 [15], and for maintaining the mitochondrial morphology [16]; in neutrophils, *GBF1* is involved in G protein-coupled receptor (GPCR)-mediated chemotaxis and superoxide production.

Loss of function (LOF) variants of *GBF1* has been hypothesized to impair motor neurons in the spinal cord, compromising Golgi apparatus function and intracellular vesicular trafficking or anterograde/retrograde cargo movement [17].

We herein report the case of two siblings with an axonal polyneuropathy carrying a new variant of the *GBF1* gene.

## 2. Materials and Methods

The present study was conducted in accordance with the recommendations of the Declaration of Helsinki. Written informed consent was obtained from the probands for their participation in this study.

Two symptomatic siblings were presented to the Neurological Department of University-Hospital Maggiore della Carità in Novara, Italy, in 2022. Their family is shown in Figure 1.

The patients underwent several analyses at our university-hospital, including peripheral blood investigations processed in the Laboratory of Biochemistry, electrophysiological studies conducted through an electroneurography (ENG) performed in the Department of Neurophysiology, and imaging studies that were processed in the Department of Radiodiagnostics and Nuclear Medicine.

The updated version of the CMT Neuropathy Score (Version 2, CMTNS) is calculated for both siblings to establish a baseline [9].

The mutational analysis of the genes was conducted by whole-exome sequencing using SureSelect Human All Exon V8 (Agilent) on the NextSeq Illumina platform. A bioinformatics analysis was performed using Illumina and wANNOVAR software. Data interpretation was performed following ACMG guidelines [7] (HGMD) using the literature data and databases such as the Human Gene Database (HGMD) and the Genome Aggregation Database (gnomAD, release 4.0.1).

## 3. Results

### 3.1. Clinical and Instrumental Assessment

#### 3.1.1. Patient II.1

A 59-year-old man presented to the Movement Disorders Clinic in February 2022 complaining of a two-year history of progressive walking difficulties, requiring bilateral support, and dizziness with subjective reduced sensitivity in his lower extremities. Symptoms began two years prior to the visit following his second COVID-19 vaccination dose.

Neurological examination revealed mild extra-pyramidal signs (asymmetrical rigidity with left-elbow trochlea after activation maneuvers and facial hypomimia, a non-reemergent bilateral postural tremor), postural instability with a positive Romberg test, and initial gait disturbance characterized by a wide-based gait and inability to perform the tandem walk test. Mild bilateral muscular atrophy in the lower limbs and hands was observed. A sensory examination showed no loss of touch, temperature or pain but a reduced vibratory sensitivity bilaterally in the lower limbs. Deep tendon reflexes (DTRs) were absent in the lower limbs and the distal radial reflex. No foot deformities or scoliosis were reported.

Medical history included hypertension (treated with β-blockers and sartans), intense smoking, and benzodiazepine use for anxiety. There was no history of diabetes, autoimmune diseases, or cancers. No exposure history (toxins, medications, alcohol or infections) was reported.

A brain CT scan was normal and did not indicate any signs of vascular parkinsonism. DAT imaging with single-photon emission computed tomography was normal. ENG revealed mixed demyelinating–axonal sensory–motor involvement- the ENG results are shown in Table 1.

First-level blood tests, including a complete blood count, were normal. Metabolic causes of neuropathy, such as diabetes, hepatic and renal dysfunctions, and thyroid disorders were ruled out. Nutritional deficiencies were tested, showing B1 hypovitaminosis, reduced folic acid, and hyperhomocysteinemia. Despite correcting these deficiencies, no clinical improvement was observed. Moreover, the clinical and electrophysiological findings, consistent with a generalized neuropathy rather than a focal one, made further imaging, as a ultrasound evaluation of peripheral nerves, unnecessary.

The CMTNS-Version 2 for the probands II.1 revealed a severe neuropathy, as shown in Table 2.

#### 3.1.2. Patient II.2

A 58-year-old woman presented in June 2022 with a five-year history of worsening paresthesia and dysesthesia in her feet, along with stiffness in the lower limbs, without reported motor symptoms.

Neurological evaluation revealed a broad-based gait with difficulty performing the tandem walk test; additionally, global and segmental strength were preserved but DTRs were absent in the lower limbs. Sensory examination showed intact tactile, thermal, and pain sensitivity, with a slightly reduced vibratory sensitivity in the lower limbs below the knees. The patient reported subjective dysesthesia in a stocking-like distribution (up to the knees bilaterally) and nocturnal cramps in the lower limbs, as well as a sensation of numbness in the lower halluxes. Bilateral cavus feet with hammer toes were noticed

Past medical history included sacral adenoma (diagnosed in 2008), Raynaud’s phenomena (since 2012), ocular Herpes Zoster infection (in 2020), hyperparathyroidism and osteoporosis treated with vitamin D supplementation (since 2022), recurrent kidney stone disease, and several pulmonary nodules with negative oncological findings. No history of diabetes or exposure to toxins, medications, alcohol, or infections was reported.

The brain and spine MRIs were unremarkable and excluded primary central nervous system involvement. ENG showed axonal sensory–motor (sensory-predominant) polyneuropathy—the ENG results are shown in Table 1. As for patient II.1, a nerve ultrasound was retained unnecessarily for the diagnostic process and was not proposed.

The blood tests, including a complete blood count, blood glucose and glycated hemoglobin, vitamin B12 and folic acid levels, as well as thyroid, liver, and kidney function tests, were unremarkable. Wilson’s disease was excluded on the basis of blood exams. Inflammatory etiologies were excluded through testing of the Erythrocyte Sedimentation Rate and C-Reactive Protein. An autoimmune panel, including anti-nuclear antibodies, Extractable Nuclear Antigen screening, anti-double-strand DNA antibodies, C3 and C4 complement fraction dosage, anti-neutrophil cytoplasmic antibodies, was unremarkable. Anti-glyadin IgG and IgM and Anti-Transglutaminase IgA and IgG antibodies were performed to rule out celiac disease. Rheumatoid factor and serum Anti-GM1, Anti-MAG, and Anti-Hu antibodies excluded neuronal-specific autoimmune diseases. Neoplastic markers were unremarkable.

Infectious etiologies, both viral and bacterial, were excluded through testing for antibodies (HIV1/2, HBV, HCV, Lyme Disease). The transthyretin gene analysis on a blood sample was negative for relevant mutations.

A lumbar puncture for the cerebrospinal fluid (CSF) analysis revealed a normal protein level (31.7 mg/dL) and cell-count with the presence of oligoclonal bands type II (CSF-isolated oligoclonal Immunoglobulines type G) and an increased K-index (36.7). The CSF analysis excluded albuminocytologic dissociation.

The CMTNS—Version 2 for the patient II.2 revealed a moderate neuropathy, as shown in Table 2.

The patient’s father (probable patient I.1) had a history of gait disturbances and required assistance with walking since middle age; he died of metastatic prostate cancer, which hindered a more precise definition of this unspecified gait disorder. Their mother passed away at 67 years due to hepatocellular carcinoma associated with HCV infection. The patients are not married and do not have children. No other affected close relatives were reported to have similar symptoms.

### 3.2. Genetic Analysis

Whole-exome sequencing revealed the presence of a new heterozygous variant in exon 10 of the *GBF1* gene (c.855-857delGTG p.Val286del), leading to an in-frame valine deletion. This variant has been classified as VUS (PM4, PM2) according to ACMG 2015 Guidelines [7]. This variant was already reported in gnomad v4.1.0 with an allele frequency of 0.00002288 in European non-Finnish. No pathogenetic variants in other lower motor neuron disease and other CMT2-associated genes were detected. Moreover, the whole-exome sequencing did not reveal any other pathogenic variant in genes associated with spastic paraplegia, Parkinson’s disease or amyotrophic lateral sclerosis.

### 3.3. Diagnosis

The siblings were diagnosed with CMT2 based on clinical features, neurophysiological findings, and the observed autosomal dominant inheritance pattern; genetic results revealed a VUS in the *GBF1* gene that was previously described in other families diagnosed with CMT2GG, albeit with a different genetic variant.

This finding allows for a possible diagnosis of CMT2GG (or AD-CMT2-Ax-*GBF1* on the basis of the new proposal classification [11], even if not definitively established, after excluding other potential differential diagnoses.

## 4. Discussion

CMT encompasses a spectrum of inherited motor and sensory neuropathies characterized by substantial clinical and genetic diversity. Our cases describe two siblings diagnosed with possible CMT2GG [13] or AD-CMT2-Ax-*GBF1* [11], a newly recognized form of adult-onset peripheral axonal neuropathy associated with mutations in the *GBF1* gene, also pointing out a suitable intra-familial clinical variability.

### 4.1. Clinical Presentation and Genetic Insights

The diagnosis of possible CMT2GG in our patients is supported by key clinical elements and suggested by genetic findings. Clinically, the observed phenotype, characterized by distal muscle weakness, sensory impairment, and axonal neuropathy, aligns closely with the typical clinical presentation of CMT2. Moreover, ENG confirmed axonal involvement, which is consistent with CMT2 pathology. Whole-exome sequencing revealed the presence of a new heterozygous variant in exon 10 of the *GBF1* gene (c.855-857delGTG p.Val286del), marking the first description of this variant in an Italian family. The Val286 residue is quite conserved throughout evolution, with the exception of chicken and zebrafish.

In our patients, no other pathogenetic variants have been identified in genes known to be associated with the CMT phenotype, nor variants associated with spastic paraplegia, motor neuron disease, or Parkinson’s disease.

Another four different heterozygous (three missense and one nonsense, namely p.Ala1137Val, p.Arg1461Gln, p.Cys982Tyr, p.Trp1175Ter) variants in different domains of the *GBF1* gene have been described in seven other cases of CMT2GG across four unrelated families [13]. Notably, the presence of a variant in the same gene, despite being classified as a VUS, in two siblings diagnosed with CMT2 (despite the high heterogeneity of the disease) allowed us to hypothesize a possible CMT2GG in this family as well. However, this discovery underscores the challenges in interpreting the clinical significance of VUS to identify rare genetic variants in neuropathies, as previously discussed [18,19]. In addition, seven different missense variants in this domain were reported in ClinVar, all of which have been classified as a VUS, although the phenotype of the subjects carrying these mutations was undefined. Altogether, these results do not exclude the possible pathogenic role of variants in this domain. Moreover, the genetic link to *GBF1* is more complex since other disease associations for variants within the GBF1 gene comprise the Bardet–Biedl Syndrome, Type 5, and cataracts.

Moreover, the extra-pyramidal phenotype of patient II.1, not determined by severe vasculopathy of basal ganglia or by functional alterations, may expand the genetic spectrum of CMT2GG described by [13], potentially implicating this gene in both peripheral neuropathies and parkinsonism, as indicated by other genome-wide association studies [20,21], where other variants in the *GBF1* gene have been associated both with Parkinson’s disease and first-degree relation to individuals with Parkinson’s disease. This is particularly relevant given the established link between mitochondrial dysregulation in genetic neuropathies involving mitofusin-1 and Parkinson’s disease associated with parkin [22]. Furthermore, late-onset parkinsonism is recognized as part of the spectrum in LRSAM1-related CMT2 [23]. Moving forward, integrating comprehensive genetic analyses with clinical data and functional assessments will be essential for accurately diagnosing and understanding the molecular basis of peripheral neuropathies.

### 4.2. Limitations and Future Directions

Our study faces several limitations. Firstly, no medical report of patient I.I was available. Additionally, the study lacks long-term follow-up data, which is essential for assessing the disease’s evolution and the long-term impact of genetic variants. An extended period for following-up is needed to capture changes in symptoms and disease progression. Confirming the pathogenicity of the *GBF1* variant with functional studies is necessary; obtaining biological samples, in addition to the DNA, could allow for assessing the presence of Golgi apparatus vesiculation and fragmentation. Currently, skin biopsies from the two patients to obtain primary fibroblast cultures are not available. Recognizing these limitations is essential for accurately interpreting our findings and highlights the need for future research to address these issues.

## 5. Conclusions

The diagnosis of possible CMT2GG (or AD-CMT2-Ax-*GBF1*) in these two siblings highlights the critical role of advanced genetic analysis and thorough clinical evaluation in identifying rare inherited neuropathies. Integrating this family into the Italian CMT Registry [2] could be beneficial for facilitating future clinical trials and potential treatments. Although gene therapies are progressing and approaching clinical trials, the unique phenotype observed presents additional diagnostic challenges and uncertainties, particularly during the recruitment phase for these clinical trials [24].

## Figures and Tables

**Figure 1 genes-15-01556-f001:**
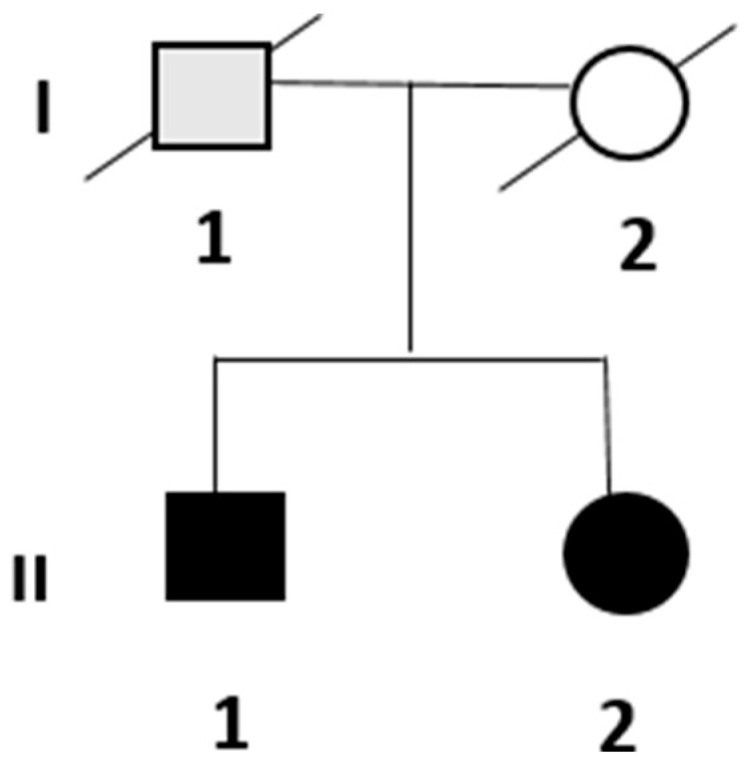
Pedigree of the family. The black-filled symbols represent symptomatic individuals for whom the genetic analysis supported the diagnosis, while the gray-filled symbol represents a potentially affected individual where a genetic analysis was not conducted.

**Table 1 genes-15-01556-t001:** Electroneurography (ENG) assessment in patient II.1 and II.2.

**Motor**	**Onset (ms)**		**Amplitude (mV)**		**Velocity (m/s)**	
	**II.1**	**II.2**	**II.1**	**II.2**	**II.1**	**II.2**
Tibial Nerve Right (at knee)	18.2	15.4	1.4	0.7	36.3	33.2
Tibial Nerve Left (at knee)	16.1	16.5	0.9	0.3	38.3	43.2
Peroneal Nerve Right (below fibula)	15.45	13.9	2.2	3.1	31.9	39.1
Peroneal Nerve Left (below fibula)	16.6	13.7	3.1	2.1	31.5	39.7
Median Right (at elbow)	-	8.2	-	4.8	-	50.0
**Sensory**	**Onset (ms)**		**Amplitude (mV)**		**Velocity (m/s)**	
	**II.1**	**II.2**	**II.1**	**II.2**	**II.1**	**II.2**
Sural Nerve Left (leg)	NR	3.2	NR	10.0	NR	46.9
Superficial Peroneal Nerve Right (lateral leg)	NR	2.5	NR	3.5	NR	32.0

The ENG assessment of both patients indicates a sensory–motor polyneuropathy in the lower limbs. NR; not recordable.

**Table 2 genes-15-01556-t002:** The CMT Neuropathy Score (Version 2) calculated in our family.

CMT Neuropathy Score (CMTNS)— Version 2 *	Patient II.1	Patient II.2
Sensory symptoms 0 = None 1 = Symptoms below or at ankle bones 3 = Up to the proximal half of the calf, including knee 4 = Above knee (above the top of the patella)	3	2
Motor symptoms (legs) 0 = None 1 = Trips, catches toes, slaps feet, Shoe inserts 2 = Ankle support or stabilization (AFOs) Foot surgery 3 = Walking aids (cane, walker) 4 = Wheelchair	2	1
Motor symptoms (arms) 0 = None 1 = Mild difficulty with buttons 2 = Severe difficulty or unable to do buttons 3 = Unable to cut most foods 4 = Proximal weakness (affect movements involving the elbow and above)	0	0
Pinprick sensibility 0 = Normal 1 = Decreased below or at ankle bones 2 = Decreased up to the distal half of the calf 3 = Decreased up to the proximal half of the calf, including knee 4 = Decreased above knee (above the top of the patella)	0	1
Vibration 0 = Normal 1 = Reduced at great toe 2 = Reduced at ankle 3 = Reduced at knee (tibial tuberosity) 4 = Absent at knee	2	2
**Total Score**	7	6

* The CMT Neuropathy Score (Version 2) by Murphy et al. [9] offers a quantitative measure of neuropathy severity and is traditionally applied to studies of CMT1A and CMT1X. A score of 0–2 is considered a mild neuropathy, with minimal impact on daily activities and limited sensory or motor involvement; 3–6 is considered a moderate neuropathy; 7–10 is considered a severe neuropathy, indicating significant disability and impact on daily activities; and 11–15 is considered a very severe neuropathy with profound functional impairment, often requiring significant assistance or aids for mobility. In this study, the CMT Neuropathy Score is utilized to provide an objective evaluation of the severity of symptoms in patients II.1 and II.2, helping to quantify the impact of CMT in these individuals.

## Data Availability

The data presented in this study are available on request from the corresponding author.

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
