# Peer review of "A Novel GBF1 Variant in a Charcot-Marie-Tooth Type 2: Insights from Familial Analysis"

_genes, 2024, doi:10.3390/genes15121556_

Round 1

Reviewer 1 Report

Comments and Suggestions for Authors

My suggestions:

1. The authors may mention a few more risk factors of CMT.

2. The authors may discuss GBF1 functions a little more in detail. 

3.  Was there any imaging performed on the patients?  Also, were any affected relatives reported (uncles, aunts, cousins, etc).

4. The authors may consider checking variants in genetic factors for other motor diseases, like spastic paraplegia, Parkinson's disease, or amyotrophic lateral sclerosis. 

5. In the discussion, the authors may discuss similar variants in GBF like (c.855-857delGTG p.Val286del (such as similar deletions, and variants in the same domain). 

Author Response

Comment 1: The authors may mention a few more risk factors of CMT.

Response 1: Thank you for your suggestion. We have reviewed the literature to identify additional risk factors associated with CMT; the disease is genetically determined with a monogenic pathway; so, an accurate family history suggestive for a symptoms that could recall a neuropathy is necessary to rule out the risk of the condition in a patient not typical for an acquired condition. 

Comment 2:  The authors may discuss GBF1 functions a little more in detail. 

Response 2: Thank you for the suggestion, I have added some additional considerations in the text regarding the function of the gene and the protein encoded in line [85; 92].

Comment 3: Was there any imaging performed on the patients?  Also, were any affected relatives reported (uncles, aunts, cousins, etc).

Response 3: A specific imaging as the nerve ultrasound has not been proposed because the neurophysiological exam was very significant for a diffuse neuropathy. In the text, we added that we haven't done the nerve ultrasound. Moreover, the two brothers are not married and do not have children; there is no information regarding similar symptoms in other relatives. The anamnestic  neurological sympthoms of their father was already reported in the manuscript.

Comment 4: The authors may consider checking variants in genetic factors for other motor diseases, like spastic paraplegia, Parkinson's disease, or amyotrophic lateral sclerosis. 

Response 4:  We have considered the genes known to be associated with motor neuron diseases and neuropathies, but no known variants have emerged. We performed  whole exome sequencing that did not reveal any other variant in genes associated to spastic paraplegia, Parkinson's disease or amyotrophic lateral sclerosis. According to reviewer suggestion, we added a sentence in the text.

Comment 5:  In the discussion, the authors may discuss similar variants in GBF like c.855-857delGTG p.Val286del (such as similar deletions, and variants in the same domain). 

Response 5: No other variant in the same domain has been associated to CMT phenotype, seven different missense variants in this domain were reported in ClinVar, all of which  have been classified as variant with Uncertain Significance, although  the phenotype of the subjects carrying these mutations was undefined. Variants in GBF1 gene  associated with neuropathies comprises missense and nonsense variants (p.Ala1137Val, p.Arg1461Gln,p. Cys982Tyr, p.Trp1175Ter) and they are located in different domanis. To date, no  in frame deletion variants have been  associated with this phenotype. Other variants described within the same gene were associated to Cataract and Bardet-Biels type 5 in accordance to MalaCard and Genecards.or .

Reviewer 2 Report

Comments and Suggestions for Authors

Authors describe the clinical and genetic data of two siblings in one family with suspected CMT2GG. I think that these case reports are very important to induce a new causative genes for CMT and probably for the new therapy for CMT.  

I think that this paper may have some value, but there are some comments.

Comments to the authors. 

Major comments: 

As authors mention in the limitation of this study, it is not clear enough that the new heterozygous variant in exon 10 of the GBF1 gene (c.855-857delGTG p.Val286del) leading to an in-frame valine deletion is causative valiant or not. Is it possible to do expression study of the valiant or the patient’s fibroblast culture as ref.13?

Is the valiant present among other species?  

Are both patients married? Do they have children? How are the children’s clinical examinations?

The clinical data are suggesting “the presence of extrapyramidal sign” in the patients. Although GBF1 is present in mouse spinal cord, is the gene expressed in basal ganglia?

Minor comments;

L58, (with a reported prevalence of 12-36% [8] > [ ( ] is necessary?

L106-110, Is the explanation of CMTNS2 grading necessary in this paragraph? Is it better in Table 1 legend?

L137, “CT scan” > “Brain CT scan”? Why not Brain MRI? Are there data of DAT scan of Patient II.2?

L140; B1 hypovitaminosis, reduced folic acid, and hyperhomocysteinemia. Despite correcting these deficiencies, no clinical improvement was observed.

L144; Nutritional deficiencies were tested, showing B1 hypovitaminosis, reduced folic acid, and hyperhomocysteinemia. Despite correcting these deficiencies, no clinical improvement was observed.

 >Repeating same things.

Table 1; Their CMTNS2 are 7 and 6 respectively. I think that the explanation of “severe neuropathy: L147” and “moderate neuropathy: L187” is not appropriate. Their CMTNS2 scores are very close.

L187; the patient II.1 > II.2

Author Response

Major comments: 

Comment 1: As authors mention in the limitation of this study, it is not clear enough that the new heterozygous variant in exon 10 of the GBF1 gene (c.855-857delGTG p.Val286del) leading to an in-frame valine deletion is causative valiant or not. Is it possible to do expression study of the valiant or the patient’s fibroblast culture as ref.13?

Response 1: Certainly the possibility of having biological samples from the two patients in addition to the DNA could allow us to see the presence of GA vesiculation and fragmentation. Unfortunately, skin biopsies from at least one of the two patients to obtain primary fibroblast cultures are not currently available.

 Comment 2: Is the valiant present among other species?  

Response 2: The Val286 residue is quite conserved throughout evolution, with the exception of chicken and zebrafish, no data are available about the presence of the same variant in other species.

 Comment 3: Are both patients married? Do they have children? How are the children’s clinical examinations?

Response 3: The patients are both not married and don’t have children; the patient II.1 lives alone and doesn’t work, and his only sister helps him in the daily activities when necessary; when we investigated their family history, no other relatives were known to have similar symptoms.

 Comment 4: The clinical data are suggesting “the presence of extrapyramidal sign” in the patients. Although GBF1 is present in mouse spinal cord, is the gene expressed in basal ganglia?

Response 4:  Thank you for your acute observation. Based on Genecards informations, the gene results expressed in spinal cord, tibial nerve and brain, in particular in amygdala, anterior cingulate cortex, caudate, cerebellar hemisphere, frontal cortex, hippocampus, hypothalamus, nucleus accumbens and putamen (basal ganglia), and substantia nigra.

Minor comments;

Comment 1: L58, (with a reported prevalence of 12-36% [8] > [ ( ] is necessary?

Response 1: Corrected.

Comment 2: L106-110, Is the explanation of CMTNS2 grading necessary in this paragraph? Is it better in Table 1 legend?

Response 2: I’ve changed the position of the CMTNS2 grading in the Table 1 legend as suggested.

Comment 3:  L137, “CT scan” > “Brain CT scan”? Why not Brain MRI? Are there data of DAT scan of Patient II.2?

Response 3: Brain CT scan was performed at first histance and, being not significant even for vasculopathy, and in absence of major cardio-vascular risk factors, a brain MRI was not mandatory for the diagnostic process.

Comment 4:  L140; B1 hypovitaminosis, reduced folic acid, and hyperhomocysteinemia. Despite correcting these deficiencies, no clinical improvement was observed. L144; Nutritional deficiencies were tested, showing B1 hypovitaminosis, reduced folic acid, and hyperhomocysteinemia. Despite correcting these deficiencies, no clinical improvement was observed. >Repeating same things.

Response 4: Thank you for your observations; as it was a refuse, I’ve corrected the right version.

Comment 5:  Table 1; Their CMTNS2 are 7 and 6 respectively. I think that the explanation of “severe neuropathy: L147” and “moderate neuropathy: L187” is not appropriate. Their CMTNS2 scores are very close.

Response 5: Thank you for your observations. I know it is a close result thus reflecting very similar severity, but we have followed the proposed classification cut-off for defining them in that way.

Comment 6: L187; the patient II.1 > II.2

Response 6: Corrected.

Round 2

Reviewer 1 Report

Comments and Suggestions for Authors

The authors fulfilled my suggestions. 

Reviewer 2 Report

Comments and Suggestions for Authors

Authors correct the manuscrips accoding to the reviewer's comments.

The weakpoint in this manuscript is no biological evidence of the mutation which is really relating to CMT type2.